# Generation of Rhesus Macaque Embryos with Expanded CAG Trinucleotide Repeats in the *Huntingtin* Gene

**DOI:** 10.3390/cells13100829

**Published:** 2024-05-13

**Authors:** Junghyun Ryu, John P. Statz, William Chan, Kiana Oyama, Maggie Custer, Martin Wienisch, Richard Chen, Carol B. Hanna, Jon D. Hennebold

**Affiliations:** 1Division of Reproductive & Developmental Sciences, Oregon National Primate Research Center, Oregon Health & Science University, Beaverton, OR 97006, USA; ryuj@ohsu.edu (J.R.); john.statz@umconnect.umt.edu (J.P.S.); william.chan@utsouthwestern.edu (W.C.); oyama@ohsu.edu (K.O.); custerma@ohsu.edu (M.C.); hannaca@ohsu.edu (C.B.H.); 2Department of Brain and Cognitive Sciences, McGovern Institute for Brain Research, Massachusetts Institute of Technology (MIT), Cambridge, MA 02139, USA; wienisch@mit.edu; 3CHDI Foundation, Princeton, NJ 08540, USA; richard.chen@chdifoundation.org; 4Assisted Reproductive Technologies Core, Oregon National Primate Research Center, Oregon Health & Science University, Beaverton, OR 97006, USA; 5Department of Obstetrics & Gynecology, Oregon Health & Science University, Portland, OR 97239, USA

**Keywords:** Huntington’s disease, rhesus macaque, CRISPR, germline editing

## Abstract

Huntington’s disease (HD) arises from expanded CAG repeats in exon 1 of the *Huntingtin* (*HTT*) gene. The resultant misfolded HTT protein accumulates within neuronal cells, negatively impacting their function and survival. Ultimately, HTT accumulation results in cell death, causing the development of HD. A nonhuman primate (NHP) HD model would provide important insight into disease development and the generation of novel therapies due to their genetic and physiological similarity to humans. For this purpose, we tested CRISPR/Cas9 and a single-stranded DNA (ssDNA) containing expanded CAG repeats in introducing an expanded CAG repeat into the *HTT* gene in rhesus macaque embryos. Analyses were conducted on arrested embryos and trophectoderm (TE) cells biopsied from blastocysts to assess the insertion of the ssDNA into the *HTT* gene. Genotyping results demonstrated that 15% of the embryos carried an expanded CAG repeat. The integration of an expanded CAG repeat region was successfully identified in five blastocysts, which were cryopreserved for NHP HD animal production. Some off-target events were observed in biopsies from the cryopreserved blastocysts. NHP embryos were successfully produced, which will help to establish an NHP HD model and, ultimately, may serve as a vital tool for better understanding HD’s pathology and developing novel treatments.

## 1. Introduction

Huntington’s disease (HD) is a genetic disorder with a global frequency of approximately 2.71 cases per 100,000 individuals [1]. The underlying mechanism of HD involves the abnormal expansion of CAG repeats in exon 1 of the *Huntingtin* (*HTT*) gene [2]. In unaffected individuals, the number of CAG repeats in the *HTT* gene typically ranges from 10 to the mid 30s. However, when the number of CAG repeats increases beyond 37, individuals are at a significantly higher risk of developing HD, with longer lengths of CAG repeats resulting in a more severe phenotype [3]. The expanded CAG repeats (~>40) in *HTT* result in an expanded polyglutamine tract within the HTT protein that leads to its accumulation [4]. Abnormal protein accumulation and aggregation of the HTT protein triggers cell death, leading to the progressive loss of neurons in various brain regions, particularly within the striatum and cerebral cortex. Furthermore, expanded CAG repeats also indirectly affect essential cellular processes for the following reasons [5,6]. Brain-derived neurotrophic factor (BDNF), a vital protein for neuronal survival and function, becomes depleted as a result of reduced BDNF transport in HD, further exacerbating neuronal demise [7]. Defective mitochondrial function also occurs, leading to diminished cellular energy production [8]. Dysregulated release of the neurotransmitter glutamate, known as glutamate excitotoxicity, also occurs in HD and results in an overload of excitatory signals that contribute to neuronal damage and cell death [9].

An animal model that recapitulates HD development and pathology is critical for developing effective therapies. Various HD models have been created to date, including in mice, pigs, sheep, and nonhuman primates (NHP) [10,11], to study the disease’s pathogenesis and develop potential treatments. NHPs such as rhesus monkeys offer a unique advantage due to their close similarities with humans in terms of their anatomy, physiology, and genetics. Moreover, rodent models have limitations in terms of replicating the human phenotype, such as limited neuronal cell death [12]. This similarity makes NHPs highly valuable for studying HD, as their disease progression and pathology would present in a manner more comparable to humans than rodents. Before CRISPR/Cas9 gene editing approaches were developed, researchers sought to create NHP HD models by utilizing the lentivirus-mediated random integration of the human *HTT* exon 1 containing 67 to 84 CAG repeats into the rhesus macaque genome [10]. The resultant transgenic NHPs demonstrated phenotypic similarities with human patients, making them valuable tools for studying HD pathogenesis and potential therapeutic interventions [13,14]. Despite the advantages of these transgenic NHP models, several challenges emerged due to the limitations of the lentivirus-mediated overexpression approach. One major concern was the random integration of the transgene into the host genome, leading to differences in the number of copies integrated and the lengths of the CAG repeat sequences across individual transgenic monkeys. As a result, each *HTT* transgenic monkey exhibited variations in the rate of disease progression and the severity of symptoms.

In this study, we employed the CRISPR/Cas9 system to assess the efficiency of directly targeting the expansion of the *HTT* CAG repeat in rhesus macaque zygotes. Alongside the CRISPR/Cas9 components, we included donor DNA in a single-strand form, which contained the desired expanded 76 CAG repeated sequence. The injected zygotes were then cultured in vitro to allow for further development. To confirm the successful integration of the CAG repeat sequence into exon 1 of the *HTT* gene by CRISPR/Cas9-mediated homology-directed repair (HDR) without altering the adjacent CAA sequence, we conducted PCR analysis and Sanger sequencing on biopsied trophectoderm (TE) cells from the blastocysts. Out of a total of 30 blastocysts, we observed an expanded CAG repeat sequence integration in 5 of them, confirming the precise genetic modification achieved through the CRISPR/Cas9 approach. Also, we assessed potential off-target loci in DNA from cryopreserved blastocysts, and two off-target events were found in sequences with mismatches of 3–4 bp relative to one of the single guide RNA (sgRNA) sequences. The future direction of this research involves producing more HD embryos using the CRISPR/Cas9 system, improving the integration of ssDNA efficiency, and subsequently establishing rhesus macaque models for HD.

## 2. Materials and Methods

### 2.1. Animals

All animals were housed at the Oregon National Primate Research Center (ONPRC), an American Association for Accreditation of Laboratory Animal Care (AAALAC)-accredited institution, and cared for in accordance with the Guide for the Care and Use of Laboratory Animals. All animal procedures conducted at this facility were performed by trained veterinary and technical staff in accordance with the Public Health Services Policy on Humane Care and Use of Laboratory Animals [15,16].

### 2.2. Design of sgRNAs and the Single-Strand DNA (ssDNA) Template

A web-based sgRNA design program (http://crispor.tefor.net/, 22 January 2024, Version 5.01) was used, and two sgRNAs were chosen, targeting *HTT* exon 1 and the adjacent intron [17]. The target sequences of the pair of gRNAs were as follows: 5′ guide: GCCTCCGGGGACTGCCTAGC; 3′ guide: GATTCCGCCATCCCCGCCGT (Figure 1).

The donor construct was assembled using standard molecular cloning techniques and contained mutant human *HTT*. Homology arms consisted of 61 base pairs upstream or downstream of the CRISPR cut sites, respectively (Appendix A). Constructs were designed prevented the cutting of the donor DNA sequence by CRISPR/Cas9 after potential ssDNA integration, and there was no modification of the amino acid sequence. The ssDNA template was produced by amplifying the assembled donor construct via PCR using a phosphorothioate-protected forward and a phosphorylated reverse primer, followed by treatment with lambda exonuclease (New England Biolabs, Ipswich, MA, USA # M0262L) to selectively digest the phosphorylated strand, leaving behind a ssDNA donor to be used for gene editing after purification. A sample of the digested product was assayed by means of gel electrophoresis to confirm the presence of a pure single band half the size of the original double-strand PCR amplicon and verified by Sanger sequencing.

### 2.3. Injection of CRISPR/Cas9 with ssDNA

A total of 18 rounds of controlled ovarian stimulations (COS) were conducted in normally cycling, breeding-age female rhesus macaques to collect mature ova for subsequent in vitro fertilization (IVF) and generation of single-cell zygotes, as previously described [18,19]. For the injection of CRISPR/Cas9 with ssDNA, a ribonucleoprotein (RNP) complex was prepared by incubating 1 µM sgRNAs (Synthego, Redwood City, CA, USA) and spCas9 protein (Integrated DNA Technologies, Coralville, Iowa, USA, #1081059) at a concentration of 112 ng/µL at room temperature for 10 min. The ssDNA was incubated in an injection buffer containing 0.1 M KCL at 80 °C for 5 min using a thermocycler and subsequently combined at room temperature with the RNP mixture at 20 ng/µL of final concentration. Microinjection was conducted in TALP-HEPES supplemented with 3 mg/mL BSA on a heated microscope stage under oil, and 50–100 pL of the injection volume was introduced into the cytoplasm or the pronucleus when visible. Injected zygotes were then cultured (BO-IVC, IVF Bioscience, Falmouth, UK) under oil in a 6%, 5%, and 89% mixture of CO_2_, O_2_, and N_2_, respectively, at 37 °C in humidified air.

### 2.4. TE Biopsy and Arrested Embryo Genotyping

The TE biopsy procedure was conducted as described previously [19]. Biopsies were performed using an objective-mounted laser (Hamilton Throne, Beverly, MA, USA). During the biopsy process, individual blastocysts were positioned in a 10 µL drop of biopsy medium (IVF Bioscience) under oil and visualized with an inverted microscope on a warmed stage. Approximately 10–15 trophectoderm cells were collected via laser dissection. Biopsied blastocysts were then cryopreserved using the global^®^ Blastocyst Fast Freeze^®^ Kit (LifeGlobal Group, Guilford, CT, USA). Embryos that had arrested at the 8-cell stage or later were exposed to acid Tyrode’s solution to remove the zona pellucida and eliminate the risk of potential sperm and granulosa transzonal projection DNA contamination during the genotyping process. Both TE biopsy and arrested embryo samples underwent whole-genome amplification (WGA) using the Repli-g Single Cell kit from Qiagen (Germantown, MD, USA). The resulting DNA was diluted at a ratio of 1:50 and served as the template for PCR amplification of the *HTT* gene target region. A pair of primers were designed on the exon 1 and the intron following exon 1 (Appendix A). For the PCR, 10 µM primers were used and the cycle parameters included an initial denaturing step at 98 °C for 3 min, followed by 98 °C for 30 s and an extension step at 72 °C for 1 min for 34 cycles, ending with a final 72 °C extension for 5 min and holding at 4 °C. Amplicons were subjected to Sanger sequencing to determine the presence of random insertion/deletion (indel) mutations or the integration of ssDNA. When multiple PCR amplicons were generated, the PCR products were ligated into the TOPO 2.1 plasmid (Invitrogen, Carlsbad, CA, USA, # 450071) and then transformed into *E. coli* (Invitrogen, #C404010). Transformed E. coli was plated on LB agar plates with ampicillin (50 µg/mL). After incubation for 16 h at 37 °C, single colonies were isolated and cultured in LB media containing 50 µg/mL ampicillin for 16 h at 37 °C, which were subsequently used for plasmid isolation (Invitrogen, K0502). Isolated plasmids were subjected to Sanger sequencing. The sequencing results were analyzed by means of BLAST against the rhesus macaque reference genome sequence to identify individual mutations.

### 2.5. Assessment of Potential False Positive Error during WGA

One nanogram of unmodified rhesus macaque genomic DNA was mixed with 1 ng of the ssDNA template and used with WGA to determine whether the presence of the ssDNA would result in the amplification of its trinucleotide repeat region due to nonspecific priming during the WGA process. To rule out another potential false-positive amplification of the expanded trinucleotide region during PCR, 30 ng of WGA-amplified genomic DNA was mixed with 10 ng of ssDNA and then used for PCR amplification of the *HTT* target region. Unmanipulated genomic DNA used for WGA was included as a negative control. Samples were then amplified using the same primers used for genotyping validation.

### 2.6. Detection of Off-Target Modifications

Unintended editing results, such as off-target events by CRISPR/Cas9 or random integration of ssDNA into high-homology sequences, were analyzed based on 5 TE biopsy samples of the cryopreserved blastocysts. To assess off-target editing, a total of three potential off-target sites were selected for each sgRNA based on the top score of the off-targets using the web-based sgRNA design program (http://crispor.tefor.net/, 22 January 2024, Version 5.01). Additionally, we identified two CAG repeat sequences in exon 1 of the rhesus macaque *androgen receptor* (*AR*) gene that were 7 and 6 repeats long. The primers were designed to amplify the 6 off-target sites and exon 1 of *AR* (Appendix A). The PCR included an initial denaturation step at 98 °C for 3 min, followed by 34 cycles of denaturation at 98 °C for 30 s, annealing at 64 °C for 30 s, and extension at 72 °C for 30 s, with a final extension at 72 °C for 5 min and holding at 4 °C. PCR products were analyzed by means of Sanger sequencing. Upon the detection of off-target events or the random insertion of ssDNA, PCR products were cloned into the TOPO 2.1 plasmid (Invitrogen, # 450071) and underwent the same procedure as utilized in embryo genotyping.

## 3. Results

### 3.1. Embryo Development after Zygote Injection

After 18 COS protocols were performed on 17 different female rhesus macaques, 438 oocytes were collected and utilized for IVF, resulting in 274 zygotes. The fertilization rate was 62.6%. All zygotes were injected with Cas9, gRNAs, and the ssDNA template with an expanded CAG repeat. The injected embryos were cultured until they were arrested or reached the blastocyst stage. Of the 274 injected zygotes, a total of 30 embryos reached the blastocyst stage, representing a 10.9% blastocyst development rate (Table 1). The remaining embryos were arrested prior to the blastocyst stage or were lysed after injection.

### 3.2. Embryo Genotyping

To explore the efficiency of ssDNA integration, DNA from 30 blastocysts (TE biopsies) and 75 randomly selected arrested embryos (>8 cell-stage) was amplified by means of PCR following WGA. The PCR products were sequenced, revealing that a total of 16 embryos had the integration of ssDNA (15.2% HDR rate) and 23 embryos possessed indel mutations on or near the sgRNA target sites (21.9% non-homologous end joining (NHEJ) editing rate). The overall editing efficiency was 37.1% (Table 1). Among the 30 blastocysts, DNA from 7 of the TE biopsies failed to yield *HTT* amplicons. The amplification of another gene on a different chromosome (*MYO7A*) was used to confirm WGA failure in these seven samples. There was no *MYO7A* PCR amplification from five samples, which indicated WGA failure. However, two samples (embryo 2413 and 2420) had *MYO7A* PCR amplicons, leading us to conclude that these two embryos may have large deletions on *HTT* exon 1 (Appendix A). Indel mutations were found in the TE biopsies of two blastocysts (Embryo 2340 and 2414), exhibiting a variety of deletions (Appendix A). Another five blastocysts were identified that possessed an expanded number of CAG repeats within exon 1 of the *HTT* gene (Figure 2A,B). Embryo 1704 exhibited a mosaic genotype that included alleles with 72 and 76 CAG repeats and a wild-type allele. Embryo 2255 possessed two different CAG repeats, where one allele had 73 repeats and another allele contained 75 repeats. Embryo 2301 was heterozygous, with one allele possessing 75 CAG repeats and another the wild-type allele. We detected 73 and 75 repeats of CAG on two different alleles from embryo 2382. Embryo 2383 had incomplete integration of ssDNA on two alleles and resulted in 21 and 23 CAG repeats. The wild-type allele was not detected in embryos 2255, 2382, and 2383 (Table 2).

During the WGA of TE biopsied samples, there is a potential risk of random priming between the genomic DNA and the homology arm sequences of any ssDNA that persists through embryo development, leading to possible false positive errors. This risk arises because the Phi 29 polymerase of the WGA kit amplifies genomic DNA through a multiple displacement amplification approach. To eliminate the concern of false positive errors, three different samples were subjected to false positive errors. The first sample was a mixture of genomic DNA and ssDNA that was subjected to WGA and then underwent PCR amplification of the target region. The second sample included a mixture of unamplified genomic DNA and ssDNA that was used for *HTT* PCR amplification. The last sample comprised WGA DNA without ssDNA that was used for *HTT* PCR amplification. From the three PCR reactions, only wild-type amplicons were detected from all combinations. This observation demonstrated that ssDNA by itself, in the absence of HDR, did not lead to WGA-associated false positive errors (Figure 3).

### 3.3. Detection of Off-Target Events from Blastocysts

No off-target events were detected in the top three off-target loci for sgRNA-1 (Appendix A). However, two off-target mutations were detected for sgRNA-2. The off-target locus sgRNA-2-1 sequence had 3 bp mismatches, and all TE biopsy samples from the blastocysts that possessed an expanded CAG repeat also carried off-target mutations in this homologous sequence in the rhesus macaque genome. For instance, embryo 2383 had a 1 bp homozygous deletion of an A, and embryo 1704 had a wild-type allele and a 1 bp A insertion. Additionally, homozygosity of the off-target events was detected for sgRNA-2-2, which had 4 bp mismatches, in embryo 2382 (Table 3 and Figure 4A,B). However, ssDNA integration was not detected at the homologous CAG repeat site of the AR exon 1 (Appendix A).

## 4. Discussion

NHPs are valuable animal models for understanding human disorders and developing novel therapeutic strategies due to their remarkable similarity to humans. Their high degree of similarity is particularly significant in the context of neurological features, rendering NHPs a suitable choice for HD modeling. For example, rhesus macaques demonstrate an exceptionally high degree of similarity to humans, with a 96% similarity in the *HTT* gene. Moreover, rhesus macaques have a relatively similar number of CAG repeats, with nine repeats, which is closer to the human sequence than other species, such as mice or pigs. In a previous study, transgenic NHPs were developed as models for HD through the utilization of the overexpression of exon 1 with extended CAG repeats of the human *HTT* gene introduced by a lentivirus. These transgenic NHPs exhibited a phenotype similar to human patients, offering valuable clinical insights into the features of HD [12]. Nevertheless, this study had a limitation in that the integration of the CAG repeat region was random with respect to the number of integrations and their location throughout the genome. The unpredictable integration site and copy number likely contributed to the generation of transgenic infants that were stillborn or had very short lifespans (1–2 days) [10].

CRISPR/Cas9 gene editing was assessed for its ability to effectively aid in the development of an HD NHP model by testing gene editing in rhesus macaque embryos. In this study, ssDNA was used as a donor template rather than double-strand DNA because ssDNA resulted in a higher integration efficiency in zebrafish embryos when compared to a double-strand DNA donor template [20]. Our approach yielded a total of 30 blastocysts from 274 fertilized embryos, indicating a blastocyst development rate of approximately 10.9% following CRISPR/Cas9 injection. It is important to note that, in comparison, the blastocyst development rate in unmanipulated embryos in the ONPRC Assisted Reproductive Technologies Core is 47% [21]. The marked decrease in the embryo development rate in CRISPR/Cas9-injected embryos could be due to the cytotoxicity of the injection materials, including sgRNA, Cas9 protein, and ssDNA. Another possible explanation for reduced blastocyst development is the occurrence of chromosomal abnormalities that result from generating double-strand DNA breaks that, in turn, lead to chromosomal instability and cell division arrest. Recent results from human and mouse embryo research revealed the generation of large deletions, chromosomal rearrangements, or ploidy issues following the injection of CRISPR/Cas9 into developing zygotes. Furthermore, these chromosomal abnormalities were shown to impact blastocyst formation, contributing to significantly reduced development rates [22,23].

The injection of CRISPR/Cas9 into zygotes resulted in double-strand break (DSB) on target sites in 37% of cases, resulting in NHEJ or HDR editing outcomes in the arrested embryos and TE biopsy samples. Overall, the DSB frequency was lower than expected based on our previous study in NHP, which resulted in a DSB ratio of up to 76% [19]. The efficiency of introducing double-strand breaks via Cas9 could be gene-dependent, whereby *HTT* accessibility or other positional effects reduce overall editing rates. Enhancing the efficiency of DSB induction using Cas9 remains crucial for editing outcomes in genetically modified NHP production. For the optimization of CRISPR/Cas9 mediated gene editing in NHPs, there are several options under consideration for improving editing rates and the introduction of an expanded CAG repeat region in the *HTT* gene, such as adjusting the injection material concentration, the timing and different forms of Cas9 [19,24,25]. In this study, the efficiency of ssDNA integration was 15.2%, which was lower than in other studies involving ssDNA injection into embryos. In zebrafish embryos, up to 90% ssDNA integration was reported, whereas 77% of pups carried a ssDNA template sequence after injecting CRISPR/Cas9 and ssDNA into mouse zygotes [20,26]. The variation in ssDNA integration efficiency might be attributed to differences in the length of the homology arm, differences in the exogenous sequence being integrated into the genome, differences in the forms of Cas9 (e.g., mRNA versus protein), the gene-dependent sgRNA editing efficiency, and/or species-specific differences. Manipulatable variables represent critical aspects of germline gene editing in NHPs that need to be optimized, not only for the successful creation of an HD monkey model but also for the production of other genetically modified NHP models.

An unanticipated outcome included the incomplete integration of ssDNA into embryo #2383, which included the integration of only 21 and 22 CAG repeats. This result indicates the incomplete insertion of ssDNA because the rhesus macaque *HTT* gene normally has nine CAG repeats. Similar observations were noted in previous embryo studies in mice and pigs, where the integration of a single-strand DNA donor was incomplete [26,27]. The reason behind this incomplete integration remains unclear; however, one potential explanation for the partial ssDNA integration could be the degradation of ssDNA due to the remaining activity of exonuclease I during the resection of broken DNA within the HDR procedure. Exonuclease I’s role involves the removal of nucleotides from linear single-stranded DNA in the 3′ to 5′ direction. Given this insight, it is assumed that the ssDNA had a complete homology arm sequence on the 5′ end. In contrast, the 21–22 CAG repeat sequence served as a homology arm on the 3′ side after exonuclease I degradation.

The effect of the type of CAG expansion on pathology is also a consideration for any HD rhesus macaques generated using the CRISPR/Cas9-HDR approach. The difference in the number of CAG repeats between the embryo groups with 72–76 repeats and those with 21–22 repeats may lead to varying degrees of severity of HD and different ages at which the HD phenotype appears, as was observed in the monkey HD models generated using a lentivirus. Therefore, when transplanting the aforementioned embryos into surrogates to create HD monkey models, it will be necessary to consider the difference in the onset timing of HD. Investigating *HTT1a* is also an important aspect of the HD monkey model. *HTT1a* is an mRNA transcript of *HTT* exon 1, containing the extended CAG repeat and part of intron 1. Its expression was reported to be associated with the number of CAG repeats and has been shown to be highly pathogenic in mouse models [28].

WGA and PCR DNA amplification for the analysis of the outcomes of the integration of ssDNA has the potential to introduce false positive errors, whereby the homology arm sequence of the ssDNA might unintentionally serve as a primer, binding to the genomic DNA during the WGA and PCR amplification processes. Thus, to assess the potential for false-positive CAG repeat expansion as a consequence of the WGA process, a mixture of genomic DNA and ssDNA was subjected to WGA. Subsequently, the flanking region of the ssDNA integration site was amplified via PCR. Another test involved subjecting the WGA of genomic DNA to PCR in the presence of the ssDNA template in the reaction mixture to assess if this combination gives rise to a false-positive CAG expansion. Assessment of the false-positive amplification of CAG repeats revealed that there were no spurious priming events with the WGA and PCR systems utilized in this study.

Unwanted editing results, such as off-target DNA editing, are a major concern in gene-edited animals, as they may result in unexpected phenotypes. In this study, two off-target sites with homology to sgRNA-2 were detected from the sgRNA 2-1 and sgRNA-2-2 sites. The off-target site, sgRNA-2-1, is located in the 5′ UTR of the *AVPR1A* gene, which could lead to an unexpected phenotype. This is particularly concerning because the main function of the UTR is the regulation of mRNA expression, and the *AVPR1A* gene is known to be associated with autism [29,30]. In contrast, there were no off-target editing events in regions homologous to the sgRNA-1 sequence. These findings imply that it is important to consider the location of the highest potential off-target events when designing sgRNAs.

This study illustrates the potential for CRISPR/Cas9 zygote injection to generate NHP embryos that can ultimately give rise to animals that are translationally relevant models for human genetic disorders. Although careful consideration of off-target editing must be taken into account, the CRISPR/Cas9 approach offers the advantage of bypassing the challenges associated with other approaches that are time-consuming and costly, such as somatic cell nuclear transfer using gene-edited cells, thus facilitating the creation of NHP disease models. Leveraging these insights could yield substantial contributions, not only regarding the development of an HD monkey model but also to the establishment of models for various other diseases. This methodology has the potential to pave the way for advancements in disease modeling that benefit from improved efficiency and accuracy, impacting the broader field of biomedical research.

## Figures and Tables

**Figure 1 cells-13-00829-f001:**
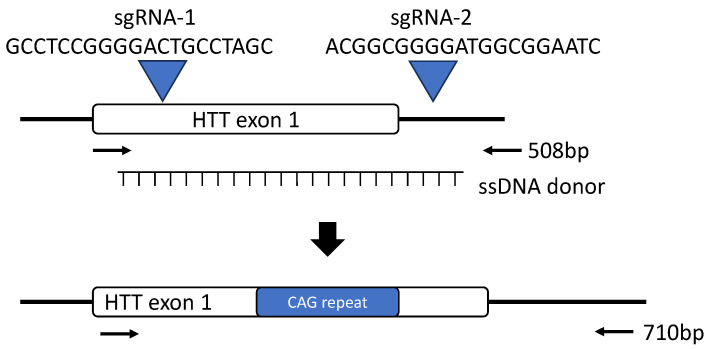
Overview of the approach employed for the integration of an expanded CAG repeat region into exon 1 of the rhesus macaque *HTT* gene. Two sgRNAs were designed to target exon 1 of *HTT* and the intron between exon 1 and 2 (arrowheads and the associated sequence). A ssDNA was synthesized that served as the donor template DNA containing an expanded CAG repeat, which was injected into rhesus macaque zygotes along with Cas9 protein complexed with the 2 sgRNAs. In cases where ssDNA integration did not occur, the PCR product should have been 508 bp. Embryos with successful ssDNA integration yielded a 710 bp PCR amplicon. Black arrows indicated PCR primers used to amplify the flanking region of *HTT* gene.

**Figure 2 cells-13-00829-f002:**
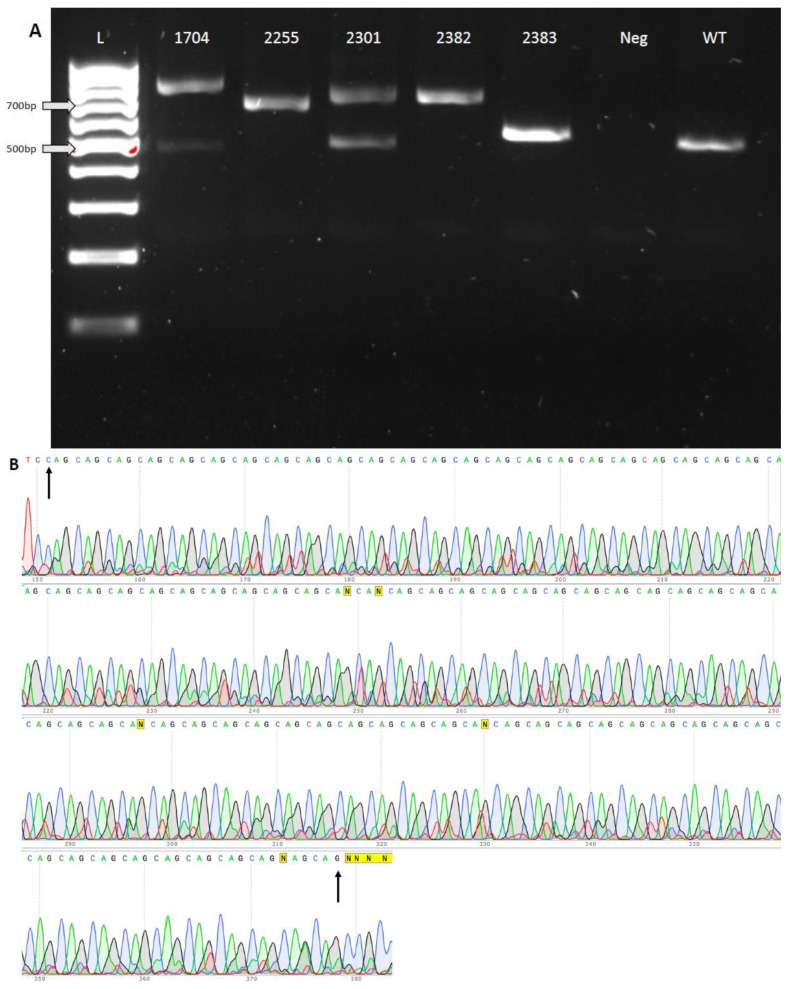
Rhesus macaque blastocysts possessing HDR-mediated expansion of the CAG repeats. (**A**) A total of 5 TE biopsy samples from blastocysts carried CAG repeats within exon 1 of the *HTT* gene that differed from wild-type or “unexpanded” CAG repeats. After the amplification of genomic DNA, putative wild-type (508 bp) or HDR (>508 bp) PCR amplicons were detected by means of gel electrophoresis. L, 100 bp ladder; Neg, negative control (no DNA); WT, PCR using genomic DNA of a rhesus macaque from the ONPRC colony. (**B**) Sanger sequencing results from embryo #1704, which possesses 76 CAG repeats, are representative of the sequencing results. CAG repeats started from bp 151 to bp 378 of the rhesus macaque *HTT* mRNA, as indicated by the black arrows.

**Figure 3 cells-13-00829-f003:**
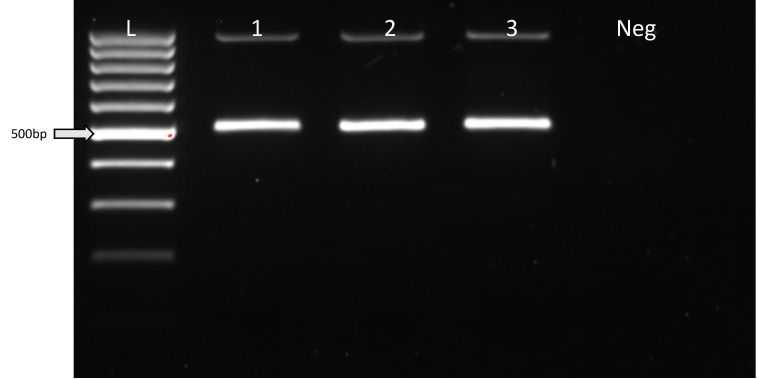
Assessment of genotyping assay accuracy. In order to assess if there is a potential for the ssDNA to lead to the false-positive amplification of CAG repeats during genotyping analysis, the following combinations of ssDNA and genomic DNA were analyzed. Sample 1 contained a mixture of genomic DNA and ssDNA, which was then subjected to WGA. Then, WGA product was used for PCR amplification of *HTT* gene using primers spanning the *HTT* gene targeting region. Sample 2 consisted of genomic DNA obtained following WGA and ssDNA that was utilized for *HTT* gene amplification by PCR. Sample 3 represented the *HTT* gene amplicon derived from the genomic DNA (WGA product). Neg, a negative PCR control that included no DNA template. L, 100 bp ladder.

**Figure 4 cells-13-00829-f004:**
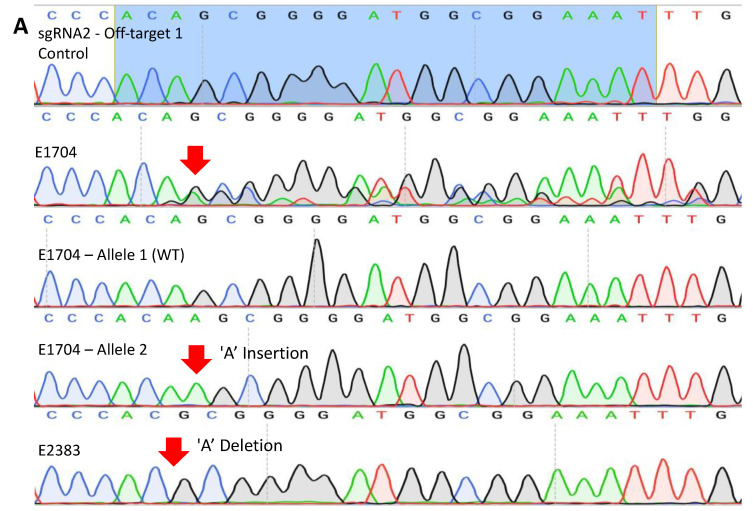
Detection of off-target events from two regions. A total of 6 potential off-target regions were PCR amplified using DNA from TE biopsies, with two sites possessing off-target editing events. (**A**) Examples of editing in a region that is homologous to the sgRNA 2 binding site, referred to as off-target-1. All 5 TE biopsy samples carried an off-target edit on the sgRNA2 off-target-1 region. (**B**) Examples of editing in a region that is homologous to the sgRNA 2 off-target site 2, referred to as off-target-2. Only one embryo TE biopsy (2382) had a homozygous one-base-pair deletion. The blue highlighted region is the off-target sequence with homology to the sgRNA-2 sequence. The top sequence represents the wild-type unedited sequence obtained from unedited control rhesus macaque DNA. Red arrows denote a mutation.

**Table 1 cells-13-00829-t001:** Embryo development and editing efficiency.

Controlled Ovarian Stimulation (COS) Cycles	# Oocytes Collected	# Fertilized ^a^	# Blastocysts ^b^	# Arrested Embryos Analyzed	# NHEJ and HDR Embryos ^c^	# NHEJ Embryos ^d^	# HDR Embryos ^e^
18	438	274 (62.6%)	30 (10.9%)	75	39 (37.1%)	23 (21.9%)	16 (15.2%)

^a^ % = number of cleavage-stage embryos/total number of oocytes that underwent IVF × 100. ^b^ % = number of blastocysts/number of fertilized oocytes × 100. ^c^ includes both arrested embryos and TE cell biopsy samples from blastocysts. ^d^ % = number of any edited embryo TE biopsies plus arrested embryos/NHEJ embryos × 100. ^e^ % = number of any edited embryo TE biopsies plus arrested embryos/HDR embryos × 100.

**Table 2 cells-13-00829-t002:** Summary of TE biopsies exhibiting an expanded CAG repeat region in the rhesus macaque *HTT* gene.

Embryo ID	CAG Repeat Insertion	Mutation
1704	Allele 1—72 repeats Allele 2—76 repeats Allele 3—WT	Mosaic mutation
2255	Allele 1—73 repeats Allele 2—75 repeats	Biallelic mutation
2031	Allele 1—75 repeats Allele 2—WT	Heterozygous mutation
2382	Allele 1—73 repeats Allele 2—75 repeats	Biallelic mutation
2383	Allele 1—21 repeats Allele 2—23 repeats	Biallelic mutation

**Table 3 cells-13-00829-t003:** Detection of off-target events from 5 cryopreserved HD embryo TE biopsies. Bold letters indicate sequence mismatches with the corresponding sgRNA. ‘+’ and ‘−’ indicate the direction of the sequences. ‘+’ means 5’-3’ and ‘−‘ shows 3’-5’.

sgRNA	Off-Target Sequence	Off-Target Score	Location	Detected Off-Target Events-TE Biopsies
sgRNA-1	1-GCCTC**A**GGG**T**ACTGCC**C**AGC (+)	CFD Off-target score: 0.532563 MIT Off-target score: 0.60	Ch10, 6,147,062–6,147,081 (Intronic)	0/5
	2-G**A**CTCC**AA**GGACTGCC**C**AGC (−)	CFD Off-target score: 0.513369 MIT Off-target score: 0.42	Ch14, 69,679–69,698, 499,553–499,572 (Intergenic)	0/5
	3-**A**CCTC**A**GGG**T**A**A**TGCCTAGC (−)	CFD Off-target score: 0.513369 MIT Off-target score: 0.42	Ch19, 6,705,306–6,705,325 (Intergenic)	0/5
sgRNA-2	1-**AT**TTCCGCCATCCCCGC**T**GT (−)	CFD Off-target score: 0.420779 MIT Off-target score: 0.66	Ch11, 62,913,702–62,913,721 (UTR)	5/5
	2-GATTCC**AT**CATCCCC**A**CC**C**T (+)	CFD Off-target score: 0.392241 MIT Off-target score: 0.06	Ch1, 96,056,543–96,056,562 (Intronic)	1/5
	3-GATTCC**C**CC**T**TCCCCGCC**AA** (−)	CFD Off-target score: 0.243730 MIT Off-target score: 0.12	Ch18, 1,238,662–1,238,681 (Intergenic)	0/5

## Data Availability

Data are contained within the article and Appendix A.

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
