# Peer review of "Generation of Rhesus Macaque Embryos with Expanded CAG Trinucleotide Repeats in the Huntingtin Gene"

_cells, 2024, doi:10.3390/cells13100829_

Round 1

Reviewer 1 Report

Comments and Suggestions for Authors

Review for Ryu et al.

Title: “Generation of rhesus macaque embryos with expanded CAG trinucleotide repeats in the Huntingtin gene.”

 Summary: This study tests a CRISPR/Cas9 editing approach to introduce an expanded CAG into the endogenous allele(s) of the macaque genome of zygotes with the aim of developing a non-human primate (NHP) model for Huntington’s disease. This is an important effort since a good NHP model has not been developed and currently the field is going from HD mice straight into patients with only safety/distribution testing in wild-type primates.

Critique:  The manuscript is quite well written, although an addition of a table summarizing the 5 embryos that were cryopreserved would make the paper much easier to digest. Also, some critical facts about macaque versus human HTT should be added, as well as clarification around whether intron1 is also expected to be edited to human sequence by the donor DNA. This is relevant since the HTT1a read-though product present in human but not all other species may be critical to pathology (although not yet proven). Some additional PCR data to assess off-target effects will strengthen the paper.

Major: There is also no mention of off-target effects of CRISPR/Cas9 other than double stranded breaks. It would strengthen the paper if the authors performed PCR for other CAG containing genes/sites to determine whether they are unintentionally expanding another site. This should be possible from existing DNA used to perform HTT and Myo7A PCR. Some discussion of off target effects and the plan for assessing this in the surviving blastocysts due to exposure to CRISPR/Cas9 RNP is necessary given that one rationale the authors provided for the study was that lentivirus could integrate randomly at sites within the genome which might create unintended effects.

Minor (Manuscript Editing):

P1, line 40: What about 36-39 cags, just to be complete?

P1, line 44: Abnormal mutant HTT ….

P1, Line 46:  “Furthermore, expanded CAG repeats also indirectly affect essential cellular processes…” vague- how??

P2, line 56: Recommend also adding sentence that mouse and other models of HD only replicate some features seen in HD patients but for instance lack neuron death.

P2, line 74: "uninterrupted 76 CAG repeat sequence followed by a single CAA"

P2, line 95: Is the intron 1 sequence of macaque highly homologous to human? Does the CRISPR edit the sequence all the way into intron1 to human?

P3, line 101: what are the “natural differences”?

P3, 102: Does this maintain amino acid sequence or are these changes in translation? If changes occur, are they conservative?

P3, 118: after heating to 80 degrees, I assume it is mixed at RT with RNP so as to not denature?

P4, Table1: adjust sizing on table so the percent in parentheses is consistently either next to or below raw number (I think is clearer below).

P5, line188: “…amplicons that revealed…” better as “amplicons, leading us to conclude that two embryos….”

P5, line 191: Please create a table of the 5 embryos that were chosen for cryopreservation and include embryo #, allele cag 1, allele cag2, and whether considered heterozygous, homozygous or mosaic, and WT or mutant. My understanding is that the rest of this paragraph refers to the 5 cryopreserved embryos?

P5, Line 193: Is 1704 one of the 5 embryos cryopreserved in the abstract? Would you use a mosaic?

P5, Line 198 What is WT CAG repeat in macaque? This is 21 and 23 which would be WT in human.

P5, line 220: “…human HTT gene introduced with lentivirus”

P5, Line 226: regarding copy number, add comment about prediction that endogenous level of mutant HTT with 76Q provide you with a phenotype useful in a timely manner for experiments.

P6, line 244: These unintended off target effects seem to be selected against since fewer blastocysts are formed. Can you assess other off-target effects? What are the chances of hitting other gens with CAG repeats and unintentionally expanding those as well? Or are your guide strands and homologous arms specific enough to rule this out?

P6, line 256: “…up to 90% ssDNA integration was reported in other species”

P6, line 267: “…which included the integration of only 21 and 22 CAG repeats.” What was the CAG repeat for normal macaque HTT?

P9, line 330: “….samples from blastocysts underwent successful amplification of the flanking region of the HTT gene”. Strike successful; it was not successful amplification for all samples based on results in B.

P9, line 337: “4 failed to amplify the MYO7A gene”  It looks like 5 samples did not amplify in supp1 fig1B? What is the expected size for the Myo7 product?

Supplementary Materials: Supplementary Figure 1. Please label sizes for at least a few markers to the ladders!

Author Response

The authors would like to thank the reviewers and the editor for the careful assessment of our manuscript “Generation of rhesus macaque embryos with expanded CAG trinucleotide repeats in the Huntingtin gene.” We appreciate the reviewers’ enthusiasm and acknowledgment of the importance of the studies included in the manuscript. The critiques are addressed below, and every effort was made to address concerns and noted weaknesses, including conducting additional experiments to better define off-target editing events.

Reviewer 1:

Summary: This study tests a CRISPR/Cas9 editing approach to introduce an expanded CAG into the endogenous allele(s) of the macaque genome of zygotes with the aim of developing a non-human primate (NHP) model for Huntington’s disease. This is an important effort since a good NHP model has not been developed and currently the field is going from HD mice straight into patients with only safety/distribution testing in wild-type primates.

We thank the reviewer for acknowledging the value of the work, particularly the importance of developing an NHP model for developing novel therapies.

Critique: The manuscript is quite well written, although an addition of a table summarizing the 5 embryos that were cryopreserved would make the paper much easier to digest. Also, some critical facts about macaque versus human HTT should be added, as well as clarification around whether intron1 is also expected to be edited to human sequence by the donor DNA. This is relevant since the HTT1a read-though product present in human but not all other species may be critical to pathology (although not yet proven).

A table (Table 2) was added as requested to provide a summary of the genotyping results for the 5 cryopreserved embryos to aid in understanding the individual HTT editing outcomes. Also, we included additional information regarding the relationship between the rhesus macaque and human HTT gene, including that the rhesus macaque has 96% similarity to the human HTT gene and rhesus macaque HTT has 9 CAG repeats in unaffected individuals while human HTT typically contains 19 CAG repeats in individuals without disease (P6 line 369-372).

The donor DNA (ssDNA) contains part of the exon 1 sequence as the 5’ homology arm, and the 3’ homology arm sequence matches the human HTT intron 1 sequence. The human intron 1 sequence differs by 6bp from the rhesus macaque sequence, resulting in a mismatch with sgRNA2 and no PAM sequence. This mismatch was used as a strategy to prevent Cas9 and sgRNA2 from generating a double-strand break after the ssDNA template is integrated into the genome. We also addressed the possibility of HTT1a expression in the edited embryos and that HTT1a should be investigated in any HD monkey model generated in the future. (P7 line 451-454)

Major: There is also no mention of off-target effects of CRISPR/Cas9 other than double stranded breaks. It would strengthen the paper if the authors performed PCR for other CAG containing genes/sites to determine whether they are unintentionally expanding another site. This should be possible from existing DNA used to perform HTT and Myo7A PCR. Some discussion of off target effects and the plan for assessing this in the surviving blastocysts due to exposure to CRISPR/Cas9 RNP is necessary given that one rationale the authors provided for the study was that lentivirus could integrate randomly at sites within the genome which might create unintended effects.

We thank the reviewer for this suggestion and agree. To address these issues, we performed additional experiments and provided detailed information in the Methods, Results, and Discussion sections. We investigated a total of 3 top-priority off-target sites for each sgRNA and found that 2 sites exhibited off-target events. Also, the androgen receptor contains a CAG repeat sequence, which potentially may lead to the integration of the ssDNA due to a conserved CAG repeat sequence. However, ssDNA integration was not detected in any of the TE biopsy samples from the cryopreserved embryos. The importance of assessing off-target editing events was noted in the Discussion section.

Minor (Manuscript Editing):

P1, line 40: What about 36-39 cags, just to be complete?

The CAG repeat range is 10 to ~35 in unaffected individuals, indicating minimal to no neuronal cell toxicity. However, more than 37 CAG repeats result in a higher risk of developing HD. This is clarified on P1, line 40.

P1, line 44: Abnormal mutant HTT ….

Additional details were included clarifying that expanded CAG repeats lead to an expanded polyglutamine tract in HTT, which in turn leads to protein aggregation and cell death (P1, line 44-45)

P2, Line 57: “Furthermore, expanded CAG repeats also indirectly affect essential cellular processes…” vague- how??

Disrupted cellular processes were described in the following sentences, which included 1) reduced BDNF production, 2) defective mitochondrial function, and 3) glutamate excitotoxicity. (P2, line 66-72)

P2, line 56: Recommend also adding sentence that mouse and other models of HD only replicate some features seen in HD patients but for instance lack neuron death.

We included additional information that mouse HD models showed less neuronal cell death. (P2. line 78-81)

P2, line 74: "uninterrupted 76 CAG repeat sequence followed by a single CAA"

It was noted that the adjacent CAA sequence after the CAG repeat was not affected. (P2, line 99)

P2, line 95: Is the intron 1 sequence of macaque highly homologous to human?

Yes, intron 1 of the rhesus monkey HTT gene is 95% identical to the human

Does the CRISPR edit the sequence all the way into intron1 to human?

No, the sgRNA targeting intron 1 is only specific for the rhesus monkey because the human has some minor differences in the same region.

P3, line 101: what are the “natural differences”?

The words ‘natural differences’ may cause confusion to the reader, and we revised the sentence. (P3 line 142-143)

P3, 102: Does this maintain amino acid sequence or are these changes in translation? If changes occur, are they conservative?

The first sgRNA is located upstream of the start codon in exon 1, while the second sgRNA is positioned in the intron. Therefore, intentionally changing the sequence of the sgRNA in the donor DNA will not affect the amino acid sequence. We have added additional information to clarify this point. (P3, line 142).

P3, 118: after heating to 80 degrees, I assume it is mixed at RT with RNP so as to not denature?

Yes, after heating at 80 degrees, the RNP and ssDNA were mixed at RT (P3, line 159).

P4, Table1: adjust sizing on table so the percent in parentheses is consistently either next to or below raw number (I think is clearer below).

The percent was moved to below the raw number (Table 1).

P5, line188: “…amplicons that revealed…” better as “amplicons, leading us to conclude that two embryos….”

Revised as recommended. (P5, line 285)

P5, line 191: Please create a table of the 5 embryos that were chosen for cryopreservation and include embryo #, allele cag 1, allele cag2, and whether considered heterozygous, homozygous or mosaic, and WT or mutant. My understanding is that the rest of this paragraph refers to the 5 cryopreserved embryos?

We addressed this issue as recommended by creating Table 2.

P5, Line 193: Is 1704 one of the 5 embryos cryopreserved in the abstract? Would you use a mosaic?

Yes, 1704 is one of the 5 HTT embryos and is mosaic. We will use 1704 embryo for embryo transfer. However, because of the mosaicism, 1704 is the lowest priority for embryo transfer.

P5, Line 198 What is WT CAG repeat in macaque? This is 21 and 23 which would be WT in human.

Rhesus macaques typically have 9 CAG repeats, which is addressed in the discussion. (P7 line 433)

21-23 times of CAG repeats in rhesus macaque may be less severe in HD and different ages onset of HD phenotype appears. (P7. line445-449)  

P5, line 220: “…human HTT gene introduced with lentivirus”

We addressed this by including additional information in the sentence (P6 line 375).

P5, Line 226: regarding copy number, add comment about prediction that endogenous level of mutant HTT with 76Q provide you with a phenotype useful in a timely manner for experiments.

We explain in the manuscript that the development of an HD disease phenotype is likely dependent on the number of repeats. (P7 line 444-451), but that those embryos with 76 repeats would produce animals that exhibit a disease phenotype.

P6, line 244: These unintended off target effects seem to be selected against since fewer blastocysts are formed. Can you assess other off-target effects? What are the chances of hitting other gens with CAG repeats and unintentionally expanding those as well? Or are your guide strands and homologous arms specific enough to rule this out?

As noted previously, we investigated off-target events, as well as the potential integration of a ssDNA into an additional CAG repeat sequence. The experiment procedure and results were addressed on the manuscript. (P4 line 218, P5 line311, and P7 linen466-474)

P6, line 256: “…up to 90% ssDNA integration was reported in other species”

In the discussion, species-specific ssDNA integration rates were included as a reference for comparison to primates.

P6, line 267: “…which included the integration of only 21 and 22 CAG repeats.” What was the CAG repeat for normal macaque HTT?

As noted above, we discussed the number of CAG repeats typically observed in the rhesus macaque HTT gene (P7 line 434).

P9, line 330: “….samples from blastocysts underwent successful amplification of the flanking region of the HTT gene”. Strike successful; it was not successful amplification for all samples based on results in B.

The word ‘successful’ was removed as requested, as the reviewer is correct; we did not obtain amplicons from a few blastocyst samples. (P11 line 595)

P9, line 337: “4 failed to amplify the MYO7A gene”  It looks like 5 samples did not amplify in supp1 fig1B? What is the expected size for the Myo7 product?

We clarified the MYO7A PCR product size in the supplementary table with primer sequences and the number of failed PCRs. (P11 line 601)

Supplementary Materials: Supplementary Figure 1. Please label sizes for at least a few markers to the ladders!

DNA ladder size indicators were included in the figure as requested.

Reviewer 2 Report

Comments and Suggestions for Authors

The present manuscript showed the generation of Huntington's disease (HD) model in rhesus macaque using CRISPR/Cas9 systems. The data was midway stage as the title showing clearly. It would be essential the embryo generated in the present study might show the pathology of HD.

How was the efficiency of generation of embryo and/or blastocysts with expanded CAG repeat in the present study compared with other methods? How much of an advantage does the present method have?

Author Response

The authors would like to thank the reviewers and the editor for the careful assessment of our manuscript “Generation of rhesus macaque embryos with expanded CAG trinucleotide repeats in the Huntingtin gene.” We appreciate the reviewers’ enthusiasm and acknowledgment of the importance of the studies included in the manuscript. The critiques are addressed below, and every effort was made to address concerns and noted weaknesses, including conducting additional experiments to better define off-target editing events.

Reviewer 2:

The present manuscript showed the generation of Huntington's disease (HD) model in rhesus macaque using CRISPR/Cas9 systems. The data was midway stage as the title showing clearly. It would be essential the embryo generated in the present study might show the pathology of HD.

How was the efficiency of generation of embryo and/or blastocysts with expanded CAG repeat in the present study compared with other methods? How much of an advantage does the present method have?

As was detailed in the Discussion section of the manuscript, there are two considerations for the utilization of the CRISPR/Cas9 system for the generation of a rhesus macaque HD monkey model production. The first relates to the observation that the blastocyst formation rate was decreased after injection of the CRISPR/Cas9 system compared with un-injected embryos (P6, line 389). Another important aspect of the current study was the low efficiency of DNA double-strand breaks (DSB) by CRISPR/Cas9 compared with our previous study (P6 line 402). In this study DSB efficiency was around 37%, but reached 76% in our previous study. The low efficiency of DSB may likely result in a reduced rate of ssDNA integration and homology-directed repair. As also noted in the Discussion, other studies demonstrated 77-90% efficiency in ssDNA integration in zebrafish and mice, which was 5- to 6-fold higher than our study.

Since somatic cell nuclear transfer efficiency is very low in rhesus macaques, injection of the CRISPR/Cas9 system into the zygote is the ideal method to generate an HD model monkey because virus-mediated integration of HTT exon1 with expended CAG has the issues that uncontrollable integration sites and copy number. Our CRISPR/Cas9 system is gradually being optimized, and currently, we have achieved up to 100% DSB at target sites. Moreover, with continuing advances in gene editing systems and efficiencies, it is likely future studies will be required to assess their ability to create NHP models of human disease, including HD.

Reviewer 3 Report

Comments and Suggestions for Authors

This is a study of using the CRISPR/Cas9 gene editing system to produce replicating zygotes of macaque monkeys (as non-Human Primates, NHP) that contain the introduced human Huntington disease (HD) gene defect (HTT) (expansion of CAG repeats in Exon 1). The authors show with whole genome amplification (WGA) of zygotes, followed by PCR assay using primers to amplify human Exon 1 of HTT, that ~10% of transformed zygotes survived (ie, replicated beyond the 8 cell stage and contained the desired 76 base pair CAG HD-causing insert.)

Throughout the study, the authors used appropriate controls, such as no DNA added, whether single-stranded DNA containing the CAG repeat would be PCR amplified in the absence of insertion into the HD gene (it did not amplify), or amplification of non-HTT genes to control for failure of WGA (this applied to 5/7 zygotes). In the other 2/7 cases, they demonstrated the presence of likely deletions introduced by the Cas9 step.

This is a sophisticated study of applying the CRISPR/Cas9 gene editing technology to introduce a disease-causing gene into a specific gene in the genome of replicating zygotes. As such, it represents a substantial improvement over the use of lentivirus transfection, which was more random. This paper lays the groundwork for developing a NHP model of HD. As such, it represents a major advance in application of gene editing technology for increased understanding of a devastating, rare, albeit classical, genetic disease whose molecular underpinnings are well known.

However, there is a major caveat that is totally separate from the elegance of this study. Should individuals with nefarious motives read and understand this paper, and we all know they exist, the authors have laid out how this model can be replicated in humans. This type of ethical dilemma has presented itself many times over the course of scientific discovery, and solutions to this dilemma elude us all. I merely wish to call this to everyone's attention and suggest that, if this paper is published, an accompanying editorial might be provided to affirm ethical responsibilities of investigators in this area.

Author Response

The authors would like to thank the reviewers and the editor for the careful assessment of our manuscript “Generation of rhesus macaque embryos with expanded CAG trinucleotide repeats in the Huntingtin gene.” We appreciate the reviewers’ enthusiasm and acknowledgment of the importance of the studies included in the manuscript. The critiques are addressed below, and every effort was made to address concerns and noted weaknesses, including conducting additional experiments to better define off-target editing events.

Reviewer 3:

This is a study of using the CRISPR/Cas9 gene editing system to produce replicating zygotes of macaque monkeys (as non-Human Primates, NHP) that contain the introduced human Huntington disease (HD) gene defect (HTT) (expansion of CAG repeats in Exon 1). The authors show with whole genome amplification (WGA) of zygotes, followed by PCR assay using primers to amplify human Exon 1 of HTT, that ~10% of transformed zygotes survived (ie, replicated beyond the 8 cell stage and contained the desired 76 base pair CAG HD-causing insert.)

Throughout the study, the authors used appropriate controls, such as no DNA added, whether single-stranded DNA containing the CAG repeat would be PCR amplified in the absence of insertion into the HD gene (it did not amplify), or amplification of non-HTT genes to control for failure of WGA (this applied to 5/7 zygotes). In the other 2/7 cases, they demonstrated the presence of likely deletions introduced by the Cas9 step.

This is a sophisticated study of applying the CRISPR/Cas9 gene editing technology to introduce a disease-causing gene into a specific gene in the genome of replicating zygotes. As such, it represents a substantial improvement over the use of lentivirus transfection, which was more random. This paper lays the groundwork for developing a NHP model of HD. As such, it represents a major advance in application of gene editing technology for increased understanding of a devastating, rare, albeit classical, genetic disease whose molecular underpinnings are well known.

We thank the reviewer for the positive comments and acknowledgement that this work serves as the foundation for future studies to develop an NHP model of HD.

However, there is a major caveat that is totally separate from the elegance of this study. Should individuals with nefarious motives read and understand this paper, and we all know they exist, the authors have laid out how this model can be replicated in humans. This type of ethical dilemma has presented itself many times over the course of scientific discovery, and solutions to this dilemma elude us all. I merely wish to call this to everyone's attention and suggest that, if this paper is published, an accompanying editorial might be provided to affirm ethical responsibilities of investigators in this area.

We thank the reviewer for this comment and share their concerns about misappropriating such information for unethical uses. The authors recently published a review that describes the technical and ethical rationale for not currently performing germline gene editing in humans.

Round 2

Reviewer 1 Report

Comments and Suggestions for Authors

Line 258 should be "No off target effects..." I think? The track changes show just an "n". Please double check for spelling errors.

I really appreciate the off-target assessments. Nicely done!

The new tables describing the embryos and the off-target events are very helpful. 

Author Response

Dear Reviewer 1,

I have attempted to locate the typo you mentioned. However, I am uncertain if line 258 contains the phrase 'No off target events...' Our assumption is that your reference was to line 312 in the track changes version, which reads 'No off-target events were detected in the top 3 ...'. If our assumption is correct, then there is no typo. When converting from the track changes to the accepted version, we could not find the typo you mentioned.

If our assumption is incorrect, please provide us with the full sentence or line number based on the track changes version. Then we can correct the typos.

Thank you.